# SequentialAttention++ for Block Sparsification: Differentiable Pruning Meets Combinatorial Optimization

**Taisuke Yasuda**[*†]
Voleon Group
yasuda.taisuke1@gmail.com

**Kyriakos Axiotis**[†]
Google Research
axiotis@google.com

**Gang Fu**[†]
Google Research
thomasfu@google.com

**MohammadHossein Bateni**
Google Research
bateni@google.com

**Vahab Mirrokni**
Google Research
mirrokni@google.com

## Abstract

Neural network pruning is a key technique towards engineering large yet scalable, interpretable, and generalizable models. Prior work on the subject has developed largely along two orthogonal directions: (1) differentiable pruning for efficiently and accurately scoring the importance of parameters, and (2) combinatorial optimization for efficiently searching over the space of sparse models. We unite the two approaches, both theoretically and empirically, to produce a coherent framework for structured neural network pruning in which differentiable pruning guides combinatorial optimization algorithms to select the most important sparse set of parameters. Theoretically, we show how many existing differentiable pruning techniques can be understood as nonconvex regularization for group sparse optimization, and prove that for a wide class of nonconvex regularizers, the global optimum is unique, group-sparse, and provably yields an approximate solution to a sparse convex optimization problem. The resulting algorithm that we propose, *SequentialAttention++*, advances the state of the art in large-scale neural network block-wise pruning tasks on the ImageNet and Criteo datasets.

## 1 Introduction

Pruning methods for neural networks [LeCun et al., 1989] replace dense weight matrices by sparse approximations, which offer improved generalization and inference efficiency in terms of storage, energy consumption, and other computational resources. In various common formulations, the problem of computing the best sparse approximation to a dense weight matrix is intractable as it generalizes the sparse linear regression problem, which is known to be NP-hard even to approximate [Natarajan, 1995, Foster et al., 2015, Gupte and Vaikuntanathan, 2021, Price et al., 2022]. Despite this fact, a wide variety of techniques have proven to be quite successful in practice. This includes magnitude pruning, $\ell_1$ regularization, greedy coordinate descent, sampling, among others.

While earlier works have focused on unstructured (i.e., entrywise) sparsity, which has been an active and fruitful area, researchers have rapidly recognized the importance of *structured* sparsity, which enforces that the sparse approximation respects certain patterns, such as block structure. These structural constraints often lead to further efficiency gains due to improved hardware utilization

---

[*]Work done while at Google Research.
[†]Corresponding author.

38th Conference on Neural Information Processing Systems (NeurIPS 2024).

[Anwar et al., 2017, Pool and Yu, 2021, Liu et al., 2022]. Our work thus focuses on developing new and improved techniques for structured sparsification of weight matrices, and in particular on block sparsification [Ma et al., 2023], which allow for a balance between performance gains from hardware utilization and reduced computation due to sparsity [Gale et al., 2023].

## 1.1 Importance scoring and combinatorial optimization

We argue that existing approaches to neural network pruning have developed along two orthogonal directions: algorithms for *importance scoring* and algorithms for *combinatorial optimization*. We roughly think of importance scoring algorithms as those that aim to select a small number of important entries (or blocks) of weight matrices, while we think of combinatorial optimization algorithms as wrapper methods that use the importance scoring algorithms as oracles to iteratively construct the desired (block) sparse weight matrix.

Among importance scoring algorithms, early popular choices have included magnitude pruning [Thimm and Fiesler, 1995, Han et al., 2015], where the magnitude of each trainable parameter serves as a proxy for its importance, as well as methods based on gradients [Karnin, 1990, Sanh et al., 2020], Hessians [LeCun et al., 1989, Hassibi et al., 1993, Singh and Alistarh, 2020, Frantar and Alistarh, 2023], and other statistics of the weights. Other works have incorporated $\ell_1$ regularization [Wen et al., 2016, Yang et al., 2019] to encourage sparsity. More recently, a class of techniques broadly termed *differentiable pruning* inspired by techniques for differentiable neural architecture search [Liu et al., 2019] have increased in popularity, where importance scores and/or soft masks are trained together with the network weights in a differentiable manner [Xiao et al., 2019, Voita et al., 2019, Kang and Han, 2020, Ramakrishnan et al., 2020, Savarese et al., 2020, Zhang et al., 2022]. Variations of this idea use the network weights themselves to represent the "importance scores", and simply use a transformation of the original network weights [Schwarz et al., 2021, Vanderschueren and Vleeschouwer, 2023, Cho et al., 2023].

As for the combinatorial optimization aspects of pruning, the use of iterative or greedy procedures has long been explored and is known to improve sparsification quality over "one-shot" uses of importance scoring algorithms [LeCun et al., 1989, Hassibi et al., 1993, Ström, 1997, Frankle and Carbin, 2019]. The work of Halabi et al. [2022] gives a theoretical justification of this observation via connections to weakly submodular optimization. Combinatorial optimization algorithms beyond greedy approaches, especially local search methods that improve sparsity patterns via local swaps such as iterative hard thresholding (IHT), have long been known in the submodular optimization literature, and have recently been shown to be extremely effective when combined with magnitude pruning [Evci et al., 2020, Peste et al., 2021, Kuznedelev et al., 2023b, Benbaki et al., 2023]. The work of Peste et al. [2021] also provides strong theoretical guarantees for their approach, *ACDC*. Similar ideas have also been termed as "neuroregeneration" in work of Liu et al. [2021].

Given these two highly fruitful approaches to the problem of pruning neural networks, it is natural to ask how recent advances in importance scoring algorithms and combinatorial optimization algorithms can work in concert. We investigate this question from both theoretical and empirical perspectives.

## 1.2 Theoretical results

We first present a theoretical investigation of differentiable pruning techniques for block sparsification when the objective function $\mathcal{L} : \mathbb{R}^n \to \mathbb{R}$ is strictly convex and differentiable. This already captures several interesting problems where block sparsification of weight matrices is desired, such as multinomial logistic regression and multiple response linear regression. We take the $n$ variables of our objective function to be partitioned into disjoint groups $\{T_i\}_{i=1}^t$ where $T_i \subseteq [n]$ and possibly have varying size. For instance, in the context of block sparsification, $\mathcal{L}$ could correspond to the multinomial logistic regression objective function with $K$ classes and $d$ features, and the $n = Kd$ variables could be partitioned into $t$ blocks $T_1, T_2, \ldots, T_t$. Furthermore, we will also consider an $\ell_2$ regularization term on the parameters $\boldsymbol{\beta}$, that is, we study variants of the problem $\min_{\boldsymbol{\beta} \in \mathbb{R}^n} \mathcal{L}(\boldsymbol{\beta}) + \lambda \|\boldsymbol{\beta}\|_2^2$. Note that explicit $\ell_2$ regularization is a standard component of machine learning architectures, and also appears *implicitly* whenever a loss function is optimized with gradient descent [Shalev-Shwartz, 2012], with the regularization parameter $\lambda$ being controlled by learning rate parameters and early stopping [Suggala et al., 2018].

Our contributions are twofold: (1) we show that a wide variety of differentiable pruning techniques can all be understood as an implementation of nonconvex regularization that generalizes the group LASSO, and (2) we show that a wide class of nonconvex regularizers give a unique 1-sparse global minimum that coincides with the unique 1-sparse global minimum of a corresponding group LASSO problem. These two results together establish that many differentiable pruning techniques work simply by identifying the *same* 1-*sparse solution as the group LASSO*. In turn, it is known that the 1-sparse solution found by the group LASSO is the variable block with the largest squared gradient [Axiotis and Yasuda, 2023], which is equivalent to the Orthogonal Matching Pursuit [Pati et al., 1993, Shalev-Shwartz et al., 2010, Liberty and Sviridenko, 2017, Elenberg et al., 2018] when applied sequentially (see Appendix B). Thus together, these results make progress towards understanding the inner workings of modern differentiable pruning methods.

### 1.2.1 Differentiable pruning as nonconvex regularization

For our first contribution, we observe that if we minimize the loss $\mathcal{L}$ with each of the variable groups $\boldsymbol{\beta}|_{T_i}$ for $i \in [t]$ replaced by a "masked" version $q(\mathbf{w}_i)\boldsymbol{\beta}|_{T_i}$, and with regularization on $\mathbf{w}$ and $\boldsymbol{\beta}$, then this problem is equivalent to another optimization problem that simply optimizes $\mathcal{L}$ with a different, and often sparsity-inducing, regularizer. A basic version of this observation already appears in works of Hoff [2017], Axiotis and Yasuda [2023], where it is shown that if the masks $q$ are just the identity, then we recover the usual group LASSO problem, that is,

$$\min_{\mathbf{w} \in \mathbb{R}^t, \boldsymbol{\beta} \in \mathbb{R}^n} \mathcal{L}(\{\mathbf{w}_i \boldsymbol{\beta}|_{T_i}\}_{i=1}^t) + \frac{\lambda}{2}\left(\|\mathbf{w}\|_2^2 + \|\boldsymbol{\beta}\|_2^2\right) = \min_{\boldsymbol{\beta} \in \mathbb{R}^n} \mathcal{L}(\boldsymbol{\beta}) + \lambda \sum_{i=1}^t \|\boldsymbol{\beta}|_{T_i}\|_2$$

where $\{\mathbf{w}_i\boldsymbol{\beta}|_{T_i}\}_{i=1}^t$ denotes the concatenation of the "masked" groups $\mathbf{w}_i\boldsymbol{\beta}|_{T_i}$ for $i \in [t]$. We generalize this observation and show that this framework also applies to other ideas popular in the differentiable pruning literature, such as applying $\ell_1$ regularization on the masks $\mathbf{w}$ to induce sparsity [Yang et al., 2019] or applying softmax-type masks such as $\exp(\mathbf{w}_i)$ [Yasuda et al., 2023]. We note that prior to our work, there was little theoretical understanding on the value of applying such techniques in the context of differentiable pruning.

We also apply similar ideas to differentiable pruning techniques that use the network weights themselves as importance scores [Schwarz et al., 2021, Cho et al., 2023]. Here, the basic observation is that if one optimizes a loss function $\mathcal{L}$ with variables $\boldsymbol{\beta}$ replaced by the (signed) entrywise square $\boldsymbol{\beta} \odot \boldsymbol{\beta}$, then this results in a "rich get richer" dynamic where large weights evolve to be larger while smaller weights are driven to zero, resulting in sparse solutions. This idea also has connections to exponentiated gradient descent which also results in sparse solutions [Vaskevicius et al., 2019, Amid and Warmuth, 2020a,b]. However, prior work only handles entrywise sparsity and does not address the question of structured pruning. We show that these ideas can also be understood in the framework of sparsity-inducing regularizers, even in the group setting. Here, we show that "masking" each of the variable groups $\boldsymbol{\beta}|_{T_i}$ by its $\ell_2$ norm $\|\boldsymbol{\beta}|_{T_i}\|_2$ gives a natural group generalization of this technique, and that this gives an optimization problem that is again equivalent to the group LASSO.

### 1.2.2 Unique sparse global minima

Our second set of contributions is to analyze the solutions of a wide class of nonconvex regularizers. We now consider the following regularized problem, where $q : \mathbb{R}_+ \to \mathbb{R}_+$ is a strictly increasing and subadditive function with $q(0) = 0$, and $\lambda > 0$ is a regularization parameter:

$$\min_{\boldsymbol{\beta} \in \mathbb{R}^n} \mathcal{L}(\boldsymbol{\beta}) + \lambda \cdot q^{-1}\left(\sum_{i=1}^t q(\|\boldsymbol{\beta}|_{T_i}\|_2)\right). \tag{1}$$

For instance, some popular choices of $q$ include the absolute value $q(x) = |x|$, $p$-th powers $q(x) = |x|^p$ for $p < 1$, or logarithmic regularizers such as $q(x) = \log(1 + x)$. In general, the class of such $q$ (strictly) contains the set of all concave functions $q$ that vanish at the origin. Note that the form of (1) slightly differs from the usual form of nonconvex regularizers, as it applies $q^{-1}$ on the sum $\sum_{i=1}^t q(\|\boldsymbol{\beta}|_{T_i}\|_2)$ rather than taking the regularizer to just be $\sum_{i=1}^t q(\|\boldsymbol{\beta}|_{T_i}\|_2)$. This does not substantially change the nature of the optimization problem as it is the Lagrangian dual for the same constraint. The main result of this section is Theorem 1.1, which relates the group $q$-regularized

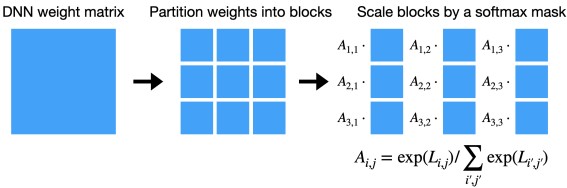

Figure 1: Differentiable pruning of weight blocks

objective (1) to the following corresponding group LASSO objective:

$$\min_{\boldsymbol{\beta} \in \mathbb{R}^n} \mathcal{L}(\boldsymbol{\beta}) + \lambda \sum_{i=1}^{t} \|\boldsymbol{\beta}|_{T_i}\|_2. \tag{2}$$

**Theorem 1.1** (Unique sparse global minima). *Let $q : \mathbb{R}_+ \to \mathbb{R}_+$ be strictly increasing, subadditive (i.e., $q(a + b) \leq q(a) + q(b)$ for $a, b \in \mathbb{R}^+$), and satisfy $q(0) = 0$. If (2) has a unique minimizer $\boldsymbol{\beta}^*$ with group sparsity at most 1, then $\boldsymbol{\beta}^*$ is also the unique minimizer for (1).*

We make several remarks about Theorem 1.1. First, we justify why the assumption of the theorem is not vacuous: that is, we explain why the group LASSO objective (2) has sparse solutions. In recent work of Axiotis and Yasuda [2023] the following are shown if $\mathcal{L}$ is strongly convex and differentiable:

- If $\lambda \geq \tau$ for $\tau = \max_{i=1}^{t} \|\nabla\mathcal{L}(0)|_{T_i}\|_2$, then (2) has a unique global minimizer at $\boldsymbol{\beta} = 0$.
- If $\lambda < \tau$ is sufficiently close to $\tau$, then (1) has a unique 1-sparse global minimizer.

Thus, when $\lambda$ is large enough, Theorem 1.1 establishes that (1) has a unique sparse global minimum.

Furthermore, Axiotis and Yasuda [2023] also show that the above global minimizer of the group LASSO problem (2) with group sparsity 1 is supported on a group $T_i$ that maximizes $\|\nabla\mathcal{L}(0)|_{T_i}\|_2$, that is, it selects the group of variables that locally provides the largest improvement in the objective function cost. Repeatedly alternating between selecting such a feature and re-optimizing over the support is an algorithm known as the *group orthogonal matching pursuit (group OMP)*, and has provable guarantees for group sparse convex optimization when $\mathcal{L}$ satisfies the restricted strong convexity and restricted smoothness properties [Axiotis and Yasuda, 2023]. It is also shown that a related local search algorithm known as *group orthogonal matching pursuit with replacement (group OMPR)* also applies in this context, which has improved guarantees.

Finally, we emphasize that it is generally difficult to establish structural results for nonconvex optimization problems, even for simple convex problems with nonconvex regularizers. Thus, we believe that our results may be of independent interest in the literature of nonconvex optimization.

### 1.3 Empirical results: SequentialAttention++

We now apply our theoretical insights of combining differentiable pruning and combinatorial optimization to develop a novel algorithm for block neural network pruning, which we call *SequentialAttention++*. SequentialAttention++ is primarily a fusion of two prior techniques: *Sequential Attention*, a feature selection technique based on differentiable pruning developed in work of Yasuda et al. [2023], and *ACDC*, which is a highly effective stochastic adaptation of the classic iterative hard thresholding (IHT) algorithm [Peste et al., 2021] from the combinatorial optimization literature.

Sequential Attention [Yasuda et al., 2023] is an algorithm for feature selection on neural networks, that introduces a softmax mask that is trained together with the neural network weights. Each of the $n$ input features is scaled by a differentiable mask $A_i = \exp(L_i)/\sum_{j=1}^{n} \exp(L_j)$ for a vector $L \in \mathbb{R}^n$ of logits. Note that our theoretical results on differentiable pruning, and in particular Lemma 2.1, suggests that this roughly corresponds to performing a log-sum regularization on the corresponding weights for these features. We first extend this to the block sparsification setting by instead scaling each block of weights to prune by a similar softmax mask (see Figure 1). Note that in this new setting, Lemma 2.1 shows that this corresponds to a *group* log-sum regularization on each of the blocks.

We then use this differentiable pruning technique as part of a local search procedure inspired by ACDC [Peste et al., 2021]. In the originally proposed ACDC algorithm, the neural network is trained

in multiple phases, where the phases alternate between a "dense" training phase and a "sparse" training phase. During the dense phases, the weights are trained in the standard way, whereas in the sparse phases, only a sparse set of weights corresponding to the top $k$ weights at the beginning of the phase (i.e., chosen by magnitude pruning) are used. The idea here is that if a suboptimal sparse support is selected during the sparse phase, then this support can be modified during the dense phase. We note that one of the weaknesses of this algorithm is the use of the weight magnitudes as a proxy for the importance of the weights, whereas improved parameter importance estimation is possible by introducing differentiable pruning techniques. Thus in our SequentialAttention++ algorithm, we modify the ACDC algorithm by training a softmax mask together with the neural network weights during the dense phase as in Figure 1, and then using the softmax mask to select a sparse support during the sparse phases. Our theoretical results establish provable guarantees for a slightly modified version of this algorithm, by showing that log-sum regularization can be integrated with a similar local search algorithm that alternates between dropping small weights from the support, selecting weights via regularization, and optimizing on the new support (see Theorem B.3 and Appendix B).

## 2 Theory

In Section 2, we present our theoretical results on differentiable pruning and local search algorithms for DNN sparsification. Missing proofs can be found in Appendix A.

### 2.1 Differentiable pruning as nonconvex regularization

In this section, we show how a wide variety of differentiable pruning techniques studied in the literature can be viewed as nonconvex regularizers. As described earlier in Section 1.2.2, we later show that nonconvex regularization can in fact be connected to provable guarantees for sparse convex optimization by implementing the orthogonal matching pursuit algorithm and its variants. Thus, together, we give the first steps towards a full theoretical analysis of many popular differentiable pruning techniques in the literature.

#### 2.1.1 Unnormalized softmax

The softmax is a popular differentiable sparsity-inducing technique, where a vector is transformed by exponentiating each entry and normalizing the result. The softmax forms the backbone of many modern ML techniques ranging from multinomial logistic regression to differentiable architecture search [Liu et al., 2019] to attention mechanisms and transformers [Vaswani et al., 2017], and thus a theoretical understanding of the softmax is critical mission for modern machine learning theory.

We take a step towards this by considering *unnormalized* softmax, which corresponds to a simple entrywise exponentiation. The unnormalized softmax is a popular alternative to the usual softmax as it still captures its sparsity-inducing properties [Amid and Warmuth, 2020a,b], while its simplicity allows for more efficient implementations. We show that, in fact, unnormalized softmax can be viewed as a type of log-sum regularization, which is a popular relaxation of the $\|\cdot\|_0$ norm that has been often considered in the machine learning and signal processing literatures [Rao and Kreutz-Delgado, 1999, Wipf and Nagarajan, 2009, Qiao et al., 2020, Tugnait, 2022, Zhou et al., 2023].

**Lemma 2.1** (Unnormalized softmax as log-sum regularization)**.**

$$\min_{\mathbf{w}\in\mathbb{R}^t, \boldsymbol{\beta}\in\mathbb{R}^n} \mathcal{L}(\{\exp(\mathbf{w}_i)\boldsymbol{\beta}|_{T_i}\}_{i=1}^t) + \lambda(\|\mathbf{w}\|_2^2 + \|\boldsymbol{\beta}\|_2^2) = \min_{\mathbf{u}\in\mathbb{R}^n} \mathcal{L}(\mathbf{u}) + \lambda\sum_{i=1}^t q(\|\mathbf{u}|_{T_i}\|_2)$$

*where* $q(a) = W(2a^2)^2/4 + W(2a^2)/2$ *and* $W$ *is the Lambert W function, i.e., the inverse of* $f(W) = We^W$.

#### 2.1.2 $\ell_1$-regularized masks

Next, we consider the idea of applying a sparsity-inducing regularization on a mask (see, e.g., the work of Yang et al. [2019]). We show that by regularizing the mask instead of the parameters themselves, the resulting optimization leads to a "more nonconvex" regularizer.

**Lemma 2.2** ($\ell_1$-regularized masks as $\ell_q$ regularization).

$$\min_{\mathbf{w}\in\mathbb{R}^t,\boldsymbol{\beta}\in\mathbb{R}^n} \mathcal{L}(\{\mathbf{w}_i\boldsymbol{\beta}|_{T_i}\}_{i=1}^t) + \lambda\big(\|\mathbf{w}\|_1 + \|\boldsymbol{\beta}\|_2^2\big) = \min_{\mathbf{u}\in\mathbb{R}^n} \mathcal{L}(\mathbf{u}) + \frac{3}{2}2^{1/3}\lambda\sum_{i=1}^t \|\mathbf{u}|_{T_i}\|_2^{2/3}$$

### 2.1.3 Powerpropagation

Finally, we study differentiable pruning techniques that use the network weights themselves as importance scores. The most straightforward implementation of this idea is to square each of the weights, as explored in works such as powerpropagation for neural networks [Schwarz et al., 2021], but more complex versions have also been considered [Cho et al., 2023]. We show how these techniques can be generalized to handle the group setting, and show how they can also be interpreted as an implementation of group sparsity-inducing regularization.

**Lemma 2.3** (Group powerpropagation as Group LASSO).

$$\min_{\mathbf{w}\in\mathbb{R}^t,\boldsymbol{\beta}\in\mathbb{R}^n} \mathcal{L}(\{\|\boldsymbol{\beta}|_{T_i}\|_2\boldsymbol{\beta}|_{T_i}\}_{i=1}^t) + \lambda\|\boldsymbol{\beta}\|_2^2 = \min_{\mathbf{u}\in\mathbb{R}^n} \mathcal{L}(\mathbf{u}) + \lambda\sum_{i=1}^t \|\mathbf{u}|_{T_i}\|_2$$

## 2.2 Unique sparse global minima

We will prove the following theorem in this section, which establishes natural conditions for which nonconvex regularization of a convex function produces a unique group-sparse global minimum. As discussed in Section 1.2.2, this theorem is the main crucial result for proving that local search algorithms give provable guarantees for sparse convex optimization.

**Theorem 1.1** (Unique sparse global minima). *Let $q : \mathbb{R}_+ \to \mathbb{R}_+$ be strictly increasing, subadditive (i.e., $q(a + b) \le q(a) + q(b)$ for $a, b \in \mathbb{R}^+$), and satisfy $q(0) = 0$. If (2) has a unique minimizer $\boldsymbol{\beta}^*$ with group sparsity at most 1, then $\boldsymbol{\beta}^*$ is also the unique minimizer for (1).*

We have the following lemma that shows that if $q$ is strictly increasing and subadditive, then the group $q$-regularization is always larger than group LASSO regularization. Thus, the group LASSO objective is always a lower bound on the $q$-regularized objective.

**Lemma 2.4.** *Let $q : \mathbb{R}_+ \to \mathbb{R}_+$ be strictly increasing and subadditive. Then,*

$$\sum_{i=1}^t \|\boldsymbol{\beta}|_{T_i}\|_2 \le q^{-1}\left(\sum_{i=1}^t q(\|\boldsymbol{\beta}|_{T_i}\|_2)\right)$$

*Proof.* Since $q$ is invertible, applying the subadditivity condition on $q(\sum_{i=1}^t \|\boldsymbol{\beta}|_{T_i}\|_2)$ and then applying $q^{-1}$ on both sides of the inequality yields the result. $\square$

Furthermore, note that for solutions $\boldsymbol{\beta}$ that have group sparsity at most 1, the group $q$-regularization has the same value as the group LASSO regularization. That is, the lower bound value can be achieved when the group sparsity is at most 1.

**Lemma 2.5.** *Let $q : \mathbb{R}_+ \to \mathbb{R}_+$ be strictly increasing and satisfy $q(0) = 0$. Then, for any $\boldsymbol{\beta} \in \mathbb{R}^n$ with group sparsity 1,*

$$\sum_{i=1}^t \|\boldsymbol{\beta}|_{T_i}\|_2 = q^{-1}\left(\sum_{i=1}^t q(\|\boldsymbol{\beta}|_{T_i}\|_2)\right).$$

*Proof.* If $\boldsymbol{\beta}$ has group sparsity at most 1, say supported on $T_j$ for some $j \in [t]$, then we have

$$q^{-1}\left(\sum_{i=1}^t q(\|\boldsymbol{\beta}|_{T_i}\|_2)\right) = q^{-1}\big(q(\|\boldsymbol{\beta}|_{T_j}\|_2)\big) = \|\boldsymbol{\beta}|_{T_j}\|_2. \qquad \square$$

Together, Lemmas 2.4 and 2.5 imply that if the group LASSO objective has a unique sparse minimum, then this is a lower bound on the optimal value that can be achieved by the $q$-regularized objective. This proves Theorem 1.1. The formal argument can be found in Appendix A.

# 3 The SequentialAttention++ algorithm

Weight magnitude is a simple and reliable importance score used to prune candidates (in our case, blocks) in a sparse optimization problem. In many cases, however, the magnitudes do not correlate very well with the true importances of the candidates. This has been observed e.g. in Axiotis and Sviridenko [2021, 2022], who showed that the magnitude pruning criterion used in the IHT algorithm is provably suboptimal even for simple sparse regression tasks, and proposed an adaptive weight decay to deal with this issue. One reason for the suboptimality of magnitude pruning is that the weights are not encouraged to be sparse during model training, leading to redundancy. Methods such as Powerpropagation [Schwarz et al., 2021] and Sequential Attention [Yasuda et al., 2023] have been proposed to address this issue by explicitly encoding a non-convexity that encourages weights to be concentrated on a sparse subset (this can be viewed as weight re-parameterization or concave regularization, as shown in Section 2).

To test the hypothesis that softmax attention weights are higher-quality importance scores, we consider one-shot block pruning based on the softmax attention scores used in Sequential Attention (see Figure 1 on how to apply it to blocks), and we compare it with block magnitude (Frobenius norm) pruning. The results in Figure 2a suggest that softmax attention scores are generally more reliable as block importance scores, especially for larger block sizes. This leads us to adopt the softmax parameterization in our algorithm.

As observed e.g. in Peste et al. [2021], one-shot pruning approaches are significantly suboptimal compared to iterative pruning approaches such as ACDC. We use a similar alternating compressed and decompressed phases approach as ACDC, but we apply it *on the softmax attention weights* instead of the block magnitudes. This establishes SequentialAttention++ as a combination between Sequential Attention and ACDC. The basic algorithm can be seen in Algorithm 2.

---

**Algorithm 1** Feed-forward layer with the basic version of SequentialAttention++ to select top $k$ parameters from a kernel $\mathbf{W}$.

    **function** FF($\mathbf{X} \in \mathbb{R}^{b \times n}$ : input batch, $t$ : training step)
        Trainable params:
        Kernel $\mathbf{W} \in \mathbb{R}^{n \times m}$, Logits $\mathbf{L} \in \mathbb{R}^{n \times m}$

        $\mathbf{A} = nm \cdot e^{\mathbf{L}} / \sum e^{\mathbf{L}}$
        $\hat{\mathbf{W}} = \mathbf{W} \odot \mathbf{A} \odot \text{Mask}(\mathbf{A}, t)$
        **return** $\mathbf{X}\hat{\mathbf{W}}$
    **end function**

---

**Algorithm 2** Attention mask. We omit SPARSIFICATION phases for simplicity.

    **function** Mask($\mathbf{A}$ : attention weights, $t$ : training step)
        Non-trainable state: mask $\in \{0, 1\}^{n \times m}$

        **if** $t$ is in a DENSE phase **then**
            mask $\leftarrow \text{top}_k(\mathbf{A})$
            **return** $\mathbf{1}_{n \times m}$
        **else if** $t$ is in a SPARSE phase **then**
            **return** mask
        **end if**
    **end function**

---

## 3.1 The SPARSIFICATION phase

One drawback of sparse/dense (compression/decompression) phases is that the dense-to-sparse transition is abrupt. Since the lowest-magnitude weights are instantly pruned, this neglects correlations between these pruned parameters. If we were to re-train the model after pruning one parameter at a time, the picture could be drastically different, since low-magnitude weights could grow (this could happen e.g. due to parameter redundancy). In fact, this effect was highlighted by Kuznedelev et al. [2023a], who devised a backward selection method based on correlations as captured by the Hessian.

Inspired by this approach, we incorporate a backward selection phase between the DENSE and SPARSE phases, which we call the SPARSIFICATION phase. In this phase, we gradually prune the least important features based on the attention weights. This gradual process allows the model to re-adjust the attention weights after some parameters are pruned. The importance of this phase is validated by ablation experiments in Appendix D.1. We use an exponential pruning schedule, to prune more aggressively in the beginning of the phase, and more carefully at the end (as we approach the desired number of candidates $k$). A comparison of the sparsity schedules of ACDC and SequentialAttention++ can be found in Figure 2b. We use the sparsity schedule $\text{sparsity}(t) = s \cdot \frac{1 - e^{-ct}}{1 - e^{-c}}$ for $t \in [0, 1]$,

| VALIDATION ACCURACY | | | | | | | |
|---|---|---|---|---|---|---|---|
| BLOCK SIZE: $8 \times 8$ | | | | BLOCK SIZE: $16 \times 16$ | | | |
| 70% | 80% | 90% | 95% | 70% | 80% | 90% | 95% |
| $-0.12$ | $-0.11$ | $+0.10$ | — | $+0.19$ | $+0.13$ | $-0.21$ | $-1.33$ |
| BLOCK SIZE: $32 \times 32$ | | | | BLOCK SIZE: $64 \times 64$ | | | |
| 68% | 78% | 88% | 92% | 58% | 66% | 74% | 79% |
| $+0.32$ | $+0.58$ | $+0.71$ | $+3.17$ | $+2.54$ | $+2.81$ | $+2.85$ | $+5.54$ |

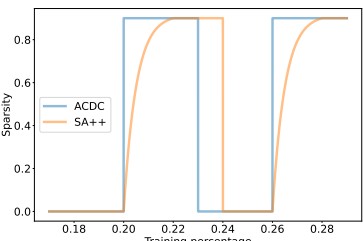

(a) Quantifying the effectiveness of magnitude vs softmax attention as block importance scores (ResNet50 on ImageNet). For different block sizes and sparsities, we show the difference between the validation accuracy (in percentage points) of a model pruned one-shot based on the softmax attention scores minus one pruned based on the block magnitudes (Frobenius norms). We train dense models for the first half of training, prune once, and then continue to train the remaining blocks for the second half of training. The experimental setup is as in Section 4.

(b) Sparsity schedules of ACDC and SequentialAttention++. ACDC uses an instant dense-to-sparse transition, while SequentialAttention++ uses an exponential sparsity schedule.

Figure 2: (a) Softmax attention vs magnitude pruning, and (b) the SPARSIFICATION phase.

where $s$ is the target sparsity. This interpolates between sparsity $0$ and $s$, and constitutes a single SPARSIFICATION phase. We choose the constant $c = 4$ (for an ablation analysis, see Appendix D.2).

## 4   Experiments

We evaluate our algorithms on sparsification tasks where a dense DNN is approximated by block-sparse counterparts, at various block sizes $B$ and sparsities $p$, where a sparsity $p$ indicates that the DNN layer will only have a $1 - p$ fraction of nonzero entries, and a block size of $B$ indicates that the nonzero entries are arranged in $B \times B$ blocks. Note that for a fixed sparsity, larger block sizes generally translate to improved efficiency due to improved hardware utilization, but also degrades quality. Block size of $1$ corresponds to unstructured pruning. Our experiments are performed on the ImageNet and Criteo datasets. More details on the setup can be found in Section C.1.

### 4.1   Baseline algorithms

We compare our SequentialAttention++ algorithm to three other representative prior algorithms for DNN pruning. The first is basic magnitude pruning, which is a popular and effective algorithm where the weights are sparsified by keeping the weights with the largest magnitude after training [Frankle and Carbin, 2019]. We use it in the block setting by keeping the largest blocks in Frobenius norm. The second algorithm is a block generalization of Powerpropagation [Schwarz et al., 2021], which combines magnitude pruning with a differentiable pruning technique where sparsity is encouraged by squaring the weights. While the original Powerpropagation algorithm did not handle the block sparsification setting, we show that multiplying each block by the Frobenius norm leads to a provable generalization (see Lemma 2.3). Finally, we consider ACDC [Peste et al., 2021], which is an adaptation of iterative hard thresholding (IHT) [Blumensath and Davies, 2009] to the setting of neural network sparsification, and has produced the state-of-the-art pruning results for ImageNet [Kuznedelev et al., 2023b]. For all algorithms and datasets, we include a fine-tuning phase at the end of training, using the pruned model, and evaluate the final pruned model on the test set.

### 4.2   Results

Our results on ImageNet are summarized in Table 1. The sparsities range over 58-95% and the block sizes over $8, 16, 32, 64$. We compare ACDC and SequentialAttention++. Our ACDC implementation closely follows the implementation in Peste et al. [2021][3]. We use the phase schedule suggested by Kuznedelev et al. [2023b] (10% dense, 7 equal SPARSE-DENSE phases where the last dense phase is extended by 5%, 15% sparse). For SequentialAttention++, we additionally replace each sparse-dense

---

[3]We sanity-checked our ACDC implementation by verifying that the accuracy of 90% unstructured global pruning matches that of the ACDC paper (75.01 vs 75.03).

Table 1: Block sparsification of ResNet50 on ImageNet. Our dense baseline validation accuracy is 76.90. The dashes are results where the algorithms diverged because of extreme sparsity. The sparsities where chosen as 70%, 80%, 90%, 95%. As seen in the table, for larger block sizes the real sparsity is lower because we are only sparsifying layers with at least 100 blocks.

| | VALIDATION ACCURACY | | | | | | | |
|---|---|---|---|---|---|---|---|---|
| | BLOCK SIZE: $8 \times 8$ | | | | BLOCK SIZE: $16 \times 16$ | | | |
| SPARSITY: | 70% | 80% | 90% | 95% | 70% | 80% | 90% | 95% |
| ACDC | 74.11 | 72.47 | 67.74 | — | 74.08 | 72.56 | 68.61 | 61.42 |
| SEQUENTIALATTENTION++ (OURS) | **74.14** | **72.90** | **69.56** | — | **74.40** | **73.50** | **69.92** | **64.27** |
| | BLOCK SIZE: $32 \times 32$ | | | | BLOCK SIZE: $64 \times 64$ | | | |
| SPARSITY: | 68% | 78% | 88% | 92% | 58% | 66% | 74% | 79% |
| ACDC | 74.40 | 72.39 | 68.96 | 63.03 | 75.18 | 74.49 | 71.95 | 67.36 |
| SEQUENTIALATTENTION++ (OURS) | **74.82** | **73.78** | **70.82** | **65.41** | **75.53** | **74.52** | **72.76** | **70.30** |

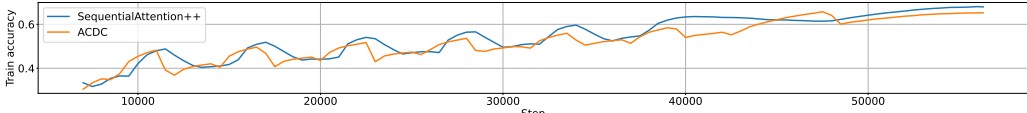

Figure 3: Training accuracy vs step on ImageNet: Comparison between ACDC and SequentialAttention++. The setting is 90% sparsity and $32 \times 32$-size blocks.

phase by a SPARSIFICATION-SPARSE-DENSE phase, as described in Section 3.1, and we replace the last of the 7 phases (including its extension) by a SPARSIFICATION phase. We use a batch size of 2048 and a maximum learning rate of 0.8.

We observe that SequentialAttention++ generally outperforms ACDC on the block sparsification task, across all different block sizes and sparsities that we tested. It should be mentioned that this comes at the cost of introducing additional trainable parameters to the model (one parameter per block). This overhead could be concerning in some applications if block size is too small (e.g., 1), in which case the model's parameters are being doubled. However, the overhead is negligible for larger (e.g., $\geq 8$) block sizes.

Our results on the Criteo dataset are presented in Table 2. The sparsities range over $p \in \{90\%, 95\%, 97\%, 98\%, 99\%\}$ and block sizes over $B \in \{5, 10, 20\}$. In this experiment, we used a schedule of 10 sparse-dense phases, in addition to a 20% initial dense phase and a final 20% sparse phase. Note that for this experiment, we used masking instead of pruning for ACDC, meaning that unselected blocks are not pruned but multiplied with an all-zero mask. We observe that SequentialAttention++ is the best performing algorithm. In fact, we notice that the gap widens with large block sizes and high sparsity, suggesting that SequentialAttention++ is a highly accurate block sparsification algorithm for large block sizes and extreme sparsities.

## 5 Conclusion

In this work, we unified, generalized, and improved prior approaches to neural network pruning via a framework which combines differentiable pruning with combinatorial optimization algorithms, in particular local search techniques. Theoretically, we gave a unified analysis of a wide class of existing techniques via a connection to nonconvex regularization, and proved novel properties about sparse convex optimization with nonconvex regularization. In particular, we established natural conditions under which nonconvex regularization yields a unique group-sparse global minimum that is supported on the group that maximizes the $\ell_2$ norm of the gradient, thus yielding provable guarantees for group sparse convex optimization. Empirically, we proposed a novel algorithm, SequentialAttention++, which outperforms prior methods on standard benchmark datasets for neural network sparsification.

We conclude with a few open directions which we believe to be interesting for future work. The first is on characterizing the nature of critical points and local minima of nonconvex-regularized convex problems. This would be a more practically useful variation on our result, which only establishes provable guarantees for the global minimizer. For our second question, we ask whether

one can theoretically establish that nonconvex regularization yields *better* optimization guarantees than the LASSO. In our work, we have only shown that the quality of solutions found by nonconvex regularization can match the LASSO for a wide variety of nonconvex regularizers, but we do not theoretically establish that this formulation is better. It would be interesting to show, e.g., that nonconvex regularizers allow for faster convergence to the sparse global minimizer.

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

# A    Missing proofs from Section 2

*Proof of Lemma 2.1.* Note first that for a fixed $a > 0$, the function $w \mapsto w^2 + a^2/\exp(2w)$ is minimized at $w$ satisfying $2w - 2a^2 \exp(-2w) = 0$, that is, $w = W(2a^2)/2$. Then, for each group $i \in [t]$, we can set $\mathbf{u}|_{T_i} = \exp(\mathbf{w}_i)\boldsymbol{\beta}|_{T_i}$ so

$$\mathbf{w}_i^2 + \|\boldsymbol{\beta}|_{T_i}\|_2^2 = \mathbf{w}_i^2 + \frac{\|\mathbf{u}|_{T_i}\|_2^2}{\exp(2\mathbf{w}_i)} \geq w^2 + w$$

where $w = W(2\|\mathbf{u}|_{T_i}\|_2^2)/2$. Summing over the groups $i \in [t]$ gives the desired result.    □

*Proof of Lemma 2.2.* Note first that for a fixed $a > 0$, the function $w \mapsto w + a^2/w^2$ is minimized at $w$ satisfying $1 - 2a^2w^{-3} = 0$, that is, $w = 2^{1/3}a^{2/3}$. Then, for each group $i \in [t]$, we can set $\mathbf{u}|_{T_i} = \mathbf{w}_i\boldsymbol{\beta}|_{T_i}$ so

$$|\mathbf{w}_i| + \|\boldsymbol{\beta}|_{T_i}\|_2^2 = |\mathbf{w}_i| + \frac{\|\mathbf{u}|_{T_i}\|_2^2}{\mathbf{w}_i^2} \geq \frac{3}{2}w$$

where $w = 2^{1/3}\|\mathbf{u}|_{T_i}\|_2^{2/3}$. Summing over the groups $i \in [t]$ gives the desired result.    □

*Proof of Lemma 2.3.* Set $\mathbf{u}|_{T_i} = \|\boldsymbol{\beta}|_{T_i}\|_2\boldsymbol{\beta}|_{T_i}$. Then,

$$\|\mathbf{u}|_{T_i}\|_2 = \|\|\boldsymbol{\beta}|_{T_i}\|_2\boldsymbol{\beta}|_{T_i}\|_2 = \|\boldsymbol{\beta}|_{T_i}\|_2^2$$

so summing over the groups gives the claimed result.    □

## A.1    Unique sparse global minima

*Proof of Theorem 1.1.* Suppose that the optimal group LASSO solution $\boldsymbol{\beta}^*$ of objective (2) has group sparsity at most 1. Then for any other solution $\boldsymbol{\beta}'$, we have that

$$\mathcal{L}(\boldsymbol{\beta}') + \lambda q^{-1}\left(\sum_{i=1}^{t} q(\|\boldsymbol{\beta}'|_{T_i}\|_2)\right)$$

$$\geq \mathcal{L}(\boldsymbol{\beta}') + \lambda \sum_{i=1}^{t}\|\boldsymbol{\beta}'|_{T_i}\|_2 \qquad \text{by Lemma 2.4}$$

$$> \mathcal{L}(\boldsymbol{\beta}^*) + \lambda \sum_{i=1}^{t}\|\boldsymbol{\beta}^*|_{T_i}\|_2 \qquad \text{by optimality}$$

$$= \mathcal{L}(\boldsymbol{\beta}^*) + \lambda q^{-1}\left(\sum_{i=1}^{t} q(\|\boldsymbol{\beta}^*|_{T_i}\|_2)\right) \qquad \text{by Lemma 2.5.}$$

Thus, $\boldsymbol{\beta}^*$ must be the unique minimizer of (1). $\qquad \square$

## B OMPR via nonconvex regularization

We show that our results from Section 2.2 together with recent work of Axiotis and Yasuda [2023] give provable guarantees for a local search algorithm based on orthogonal matching pursuit with replacement using nonconvex regularization.

We first introduce some definitions needed to state our result.

**Definition B.1.** Let $T_i \subseteq [n]$ for $i \in [t]$ form a partition of $[n]$. Then, we define
$$\|\boldsymbol{\beta}\|_{\text{group}} := |\{i \in [t] : \boldsymbol{\beta}|_{T_i} \neq 0\}|.$$

**Definition B.2** (Restricted strong convexity and smoothness). Let $\mathcal{L} : \mathbb{R}^n \to \mathbb{R}$ be differentiable. Let $T_i \subseteq [n]$ for $i \in [t]$ form a partition of $[n]$. Then, $l$ is $\mu_s$-*restricted strongly convex at group sparsity* $s$ if for any $\boldsymbol{\beta} \in \mathbb{R}^n$ and $\boldsymbol{\Delta} \in \mathbb{R}^n$ with $\|\boldsymbol{\Delta}\|_{\text{group}} \leq s$,
$$\mathcal{L}(\boldsymbol{\beta} + \boldsymbol{\Delta}) - \mathcal{L}(\boldsymbol{\beta}) - \langle \nabla \mathcal{L}(\boldsymbol{\beta}), \boldsymbol{\Delta} \rangle \geq \frac{\mu_s}{2}\|\boldsymbol{\Delta}\|_2^2,$$

and $L_s$-*restricted smooth at group sparsity* $s$ if for any $\boldsymbol{\beta} \in \mathbb{R}^n$ and $\boldsymbol{\Delta} \in \mathbb{R}^n$ with $\|\boldsymbol{\Delta}\|_{\text{group}} \leq s$,
$$\mathcal{L}(\boldsymbol{\beta} + \boldsymbol{\Delta}) - \mathcal{L}(\boldsymbol{\beta}) - \langle \nabla \mathcal{L}(\boldsymbol{\beta}), \boldsymbol{\Delta} \rangle \leq \frac{L_s}{2}\|\boldsymbol{\Delta}\|_2^2.$$

We will now obtain provable guarantees for Algorithm 3 in Theorem B.3.

---

**Algorithm 3** OMPR via nonconvex regularization

---

Initialize $S$ arbitrarily such that $|S| = k'$
**for** $i = 1, \ldots, R$ **do**
  Let
$$\hat{\boldsymbol{\beta}} = \arg\min_{\boldsymbol{\beta} \in \mathbb{R}^n} \mathcal{L}(\boldsymbol{\beta}) + \lambda \cdot q^{-1}\left(\sum_{i \notin S} q(\|\boldsymbol{\beta}|_{T_i}\|_2)\right)$$

  for $\lambda$ sufficiently large
  Let $i \notin S$ be the group maximizing $\hat{\boldsymbol{\beta}}|_{T_i}$ and $j \in S$ be the group minimizing $\|\boldsymbol{\beta}\|_2|_{T_j}$
  $S \leftarrow S \cup \{i\} \setminus \{j\}$
**end for**

---

**Theorem B.3** (OMPR via nonconvex regularization). *Let* $q : \mathbb{R}_+ \to \mathbb{R}_+$ *be strictly increasing, subadditive, and* $0$ *at the origin. After* $R$ *iterations of Algorithm 3 with* $k' \geq k\left(\frac{L_2^2}{\mu_{k+k'}^2} + 1\right)$, *for*

$$R \geq k \cdot \frac{L_2}{\mu_{k+k'}} \log \frac{\mathcal{L}(\boldsymbol{\beta}^{(0)}) - \mathcal{L}(\boldsymbol{\beta}^*)}{\varepsilon},$$

*then* $\hat{\boldsymbol{\beta}}$ *has group sparsity* $\|\boldsymbol{\beta}^\infty\|_{\text{group}} \leq k'$ *and satisfies*
$$\mathcal{L}(\boldsymbol{\beta}^\infty) \leq \mathcal{L}(\boldsymbol{\beta}^*) + \varepsilon,$$

*where* $\mu_{k+k'}$ *is a lower bound on the restricted strong convexity constant of* $l$ *at group sparsity* $k + k'$ *and* $L_2$ *is an upper bound on the restricted smoothness constant of* $l$ *at group sparsity* $2$ *(see Definition B.2).*

*Proof.* By Theorem 1.1, if the optimization problem in Algorithm 3 with $q$ replaced by the absolute value function has a unique minimizer with group sparsity at most 1, then $\hat{\boldsymbol{\beta}}$ is a unique global minimizer with group sparsity at most 1, and coincides with this Group LASSO solution. Lemma 3.2 of Axiotis and Yasuda [2023] then establishes that this solution is supported on the group that maximizes the $\ell_2$ norm of the gradient, which in turn implies Theorem B.3 via guarantees for the group orthogonal matching pursuit with replacement algorithm (Corollary A.10 of Axiotis and Yasuda [2023]). $\qquad \square$

Table 2: Block sparsification on Criteo. The validation losses are an average of three runs. Our dense baseline validation loss is $0.4489$.

|  | VALIDATION LOSS | | |
| --- | --- | --- | --- |
| SPARSITY: 90% | BLOCK SIZE: 5 | BLOCK SIZE: 10 | BLOCK SIZE: 20 |
| MAGNITUDE | 0.4523 | 0.4693 | 0.4923 |
| POWERPROPAGATION | 0.4521 | 0.4572 | 0.4920 |
| ACDC | 0.4517 | 0.4580 | 0.4829 |
| SEQUENTIALATTENTION++ (OURS) | **0.4515** | **0.4535** | **0.4596** |
| SPARSITY: 95% | BLOCK SIZE: 5 | BLOCK SIZE: 10 | BLOCK SIZE: 20 |
| MAGNITUDE | 0.4586 | 0.4892 | 0.4998 |
| POWERPROPAGATION | 0.4547 | 0.4768 | 0.4946 |
| ACDC | 0.4547 | 0.4754 | 0.4961 |
| SEQUENTIALATTENTION++ (OURS) | **0.4540** | **0.4595** | **0.4715** |
| SPARSITY: 97% | BLOCK SIZE: 5 | BLOCK SIZE: 10 | BLOCK SIZE: 20 |
| MAGNITUDE | 0.4656 | 0.5004 | 0.5079 |
| POWERPROPAGATION | 0.4587 | 0.5061 | 0.5093 |
| ACDC | 0.4606 | 0.4936 | 0.5056 |
| SEQUENTIALATTENTION++ (OURS) | **0.4570** | **0.4708** | **0.4865** |
| SPARSITY: 98% | BLOCK SIZE: 5 | BLOCK SIZE: 10 | BLOCK SIZE: 20 |
| MAGNITUDE | 0.4717 | 0.5145 | 0.5447 |
| POWERPROPAGATION | 0.4622 | 0.5158 | 0.5379 |
| ACDC | 0.4692 | 0.4929 | 0.5184 |
| SEQUENTIALATTENTION++ (OURS) | **0.4601** | **0.4904** | **0.5162** |
| SPARSITY: 99% | BLOCK SIZE: 5 | BLOCK SIZE: 10 | BLOCK SIZE: 20 |
| MAGNITUDE | 0.4881 | 0.5376 | 0.5482 |
| POWERPROPAGATION | 0.5017 | 0.5295 | 0.5425 |
| ACDC | 0.5050 | 0.5153 | 0.5427 |
| SEQUENTIALATTENTION++ (OURS) | **0.4803** | **0.5068** | **0.5253** |

# C  Additional details on experiments

## C.1  Experimental setup

**ImageNet [Deng et al., 2009].**    ImageNet is the most widely used vision dataset and is considered as the de facto benchmark in the neural network pruning literature, culminating in the state of the art results in Kuznedelev et al. [2023b]. We use ResNet50 and a standard training setup (90 epochs, SGD with cosine learning rate and momentum, weight decay). We reshape the 4-dimensional ($H \times W \times C_{\text{in}} \times C_{\text{out}}$) kernel tensors used in convolutional layers to 2D matrices of shape $HWC_{\text{in}} \times C_{\text{out}}$, which define the 2D block candidates for pruning. We prune all layers uniformly, except for layers with $< 100$ blocks, which we do not prune at all, to avoid degeneracy at high sparsities.

**Criteo [Diemert et al., 2017].**    Criteo is a standard public dataset for the clickthrough rate (CTR) prediction task, which consists of 33M training examples with 13 numerical and 26 categorical features. The model we sparsify is a standard fully connected DNN with three 400-width layers and an additional embedding layer to transform each input feature into an embedding vector of size 10 (for a total embedding width of 390). We note that a simple MLP is often a fairly competitive model for this task [Naumov et al., 2019]. We prune the first dense layer after the embedding layer. We use Adam optimizer with a learning rate that decays exponentially from $2 \cdot 10^{-2}$ to $3 \cdot 10^{-4}$. We train to minimize the cross-entropy loss for 25 epochs with a batch size of 32768.

## C.2  Block sparsification results on Criteo

We give our block sparsification results on the Criteo dataset in Table 2.

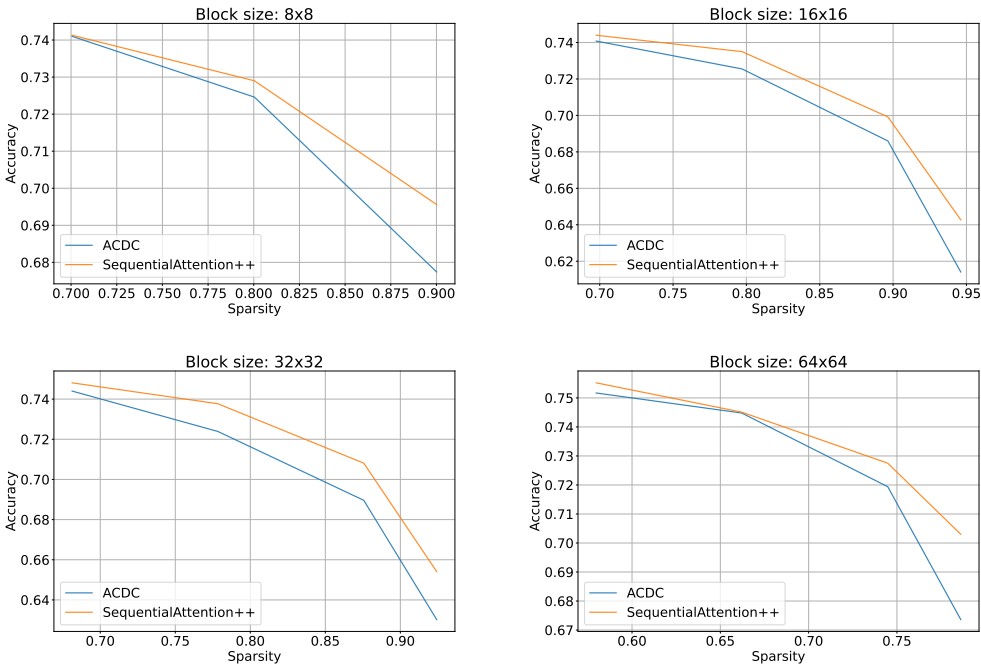

Figure 4: Block sparsification on Imagenet.

## C.3 Additional tricks

In addition to the basic algorithm described in Section 3, our implementation of SequentialAttention++ incorporates several other ingredients for improved empirical performance. First, we confirm the observation of Peste et al. [2021] that resetting the optimizer between each phase of SequentialAttention++ is crucial for good performance. We note that this is also suggested by our theoretical results (Theorem B.3), which suggests that each of the dense and sparse phases should be thought of as a separate optimization problem that is solved independently. Similarly to Kuznedelev et al. [2023b], we also observe that weight decay significantly boosts performance, even when applied to the attention logits.

Second, we observe that pruning each layer of the network separately performs better than a global pruning algorithm which attempts to prune all layers at once. We suggest that this may be the case due to "bottlenecking" behavior, where a global pruning algorithm may choose to almost completely eliminate a layer which may destroy the connectivity of the neural network. While this is not the case when pruning individual parameters, pruning large blocks can easily eliminate a layer. We use uniform sparsity across layers, but choose not to sparsify layers containing less than 100 blocks. This is because layers have greatly varying sizes, and want to avoid a sharp quality drop from overpruning smaller layers, which was observed in experiments. Finally, we clip attention weights to the range $[n \cdot \text{density}, n/\text{density}]$ to avoid them becoming too small or too large.

## C.4 Additional results

We provide additional plots for our experiments in Figures 4 and 5. In Figure 4, we plot tradeoffs between the validation accuracy and weight matrix sparsity for SequentialAttention++ and ACDC Peste et al. [2021]. In Figure 5, we plot tradeoffs between the validation loss and AUC against weight matrix sparsity for SequentialAttention++ and our three baseline algorithms of Magnitude Pruning, Powerpropagation Schwarz et al. [2021], and ACDC Peste et al. [2021].

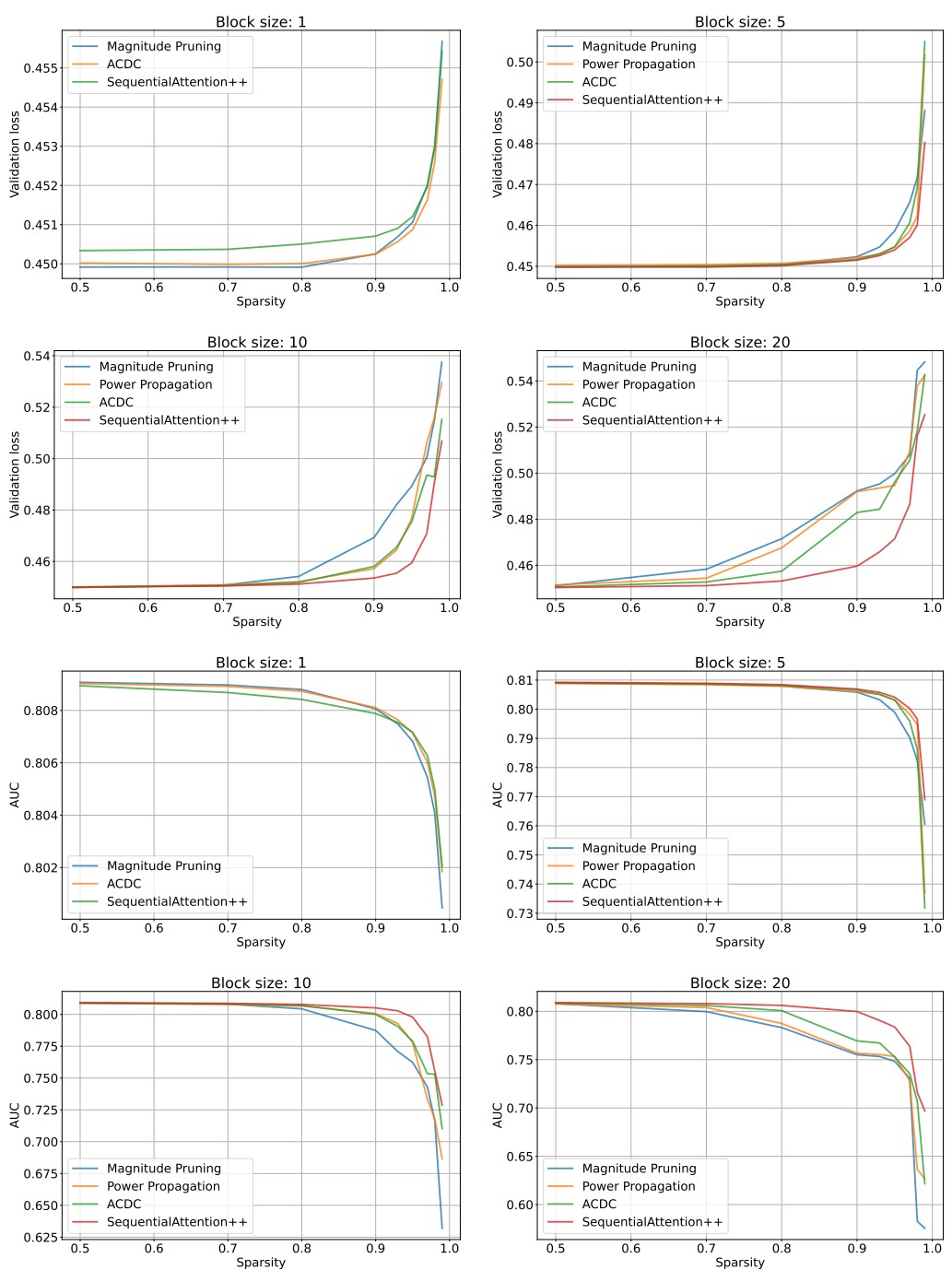

Figure 5: Block sparsification on Criteo. There are no Powerpropagation results for block size 1 because the algorithm diverged.

# D Ablations

## D.1 Importance of the SPARSIFICATION phase.

We perform experiments to study the effect of the SPARSIFICATION phase, as described in Section 3.1, to the final accuracy. To that end, we remove the SPARSIFICATION phase and only apply alternating

Table 3: Removing the SPARSIFICATION phase from SequentialAttention++. The results show validation accuracy for training block-sparse ResNet50 on ImageNet. We use the same sparsities as in Table 1.

| | VALIDATION ACCURACY | | | | | | | |
|---|---|---|---|---|---|---|---|---|
| | BLOCK SIZE: $8 \times 8$ | | | | BLOCK SIZE: $16 \times 16$ | | | |
| SPARSITY: | 70% | 80% | 90% | 95% | 70% | 80% | 90% | 95% |
| VALIDATION ACCURACY | — | 0.61922 | 0.61405 | 0.69678 | 0.68152 | 0.68658 | 0.70079 | 0.72845 |
| DIFF FROM BASELINE | — | $-0.0235$ | $-0.04006$ | $-0.00624$ | $-0.01408$ | $-0.01262$ | $-0.00738$ | $+0.00089$ |
| | BLOCK SIZE: $32 \times 32$ | | | | BLOCK SIZE: $64 \times 64$ | | | |
| SPARSITY: | 68% | 78% | 88% | 92% | 58% | 66% | 74% | 79% |
| VALIDATION ACCURACY | 0.72333 | 0.72666 | 0.73346 | 0.7432 | 0.74194 | 0.74099 | 0.74268 | 0.75104 |
| DIFF FROM BASELINE | $-0.00569$ | $-0.00837$ | $-0.00429$ | $-0.00196$ | $+0.00059$ | $-0.00301$ | $-0.00547$ | $-0.00421$ |

Table 4: Modifying the exponent constant in the schedule of the SPARSIFICATION phase. Block-sparse training of ResNet50 on ImageNet for 90% sparsity.

| | VALIDATION ACCURACY | | | |
|---|---|---|---|---|
| BLOCK SIZE | $8 \times 8$ | $16 \times 16$ | $32 \times 32$ | $64 \times 64$ |
| $c = 2$ | 0.69403 | 0.69613 | 0.70614 | 0.72264 |
| $c = 4$ | 0.6956 | 0.6992 | 0.70817 | 0.72756 |
| $c = 8$ | 0.69202 | 0.70036 | 0.70976 | 0.72614 |

DENSE and SPARSE phases, each of equal duration. The final phase before the last fine-tuning is now a DENSE phase.

The results in Table 3 show that, on average over different block sizes and densities, removing the SPARSIFICATION phase decreases validation accuracy by 0.009, or 0.9 percentage points. We conclude that the SPARSIFICATION phase is an important feature of SequentialAttention++.

### D.2 Choice of the SPARSIFICATION exponent.

In this section, we try different values of the constant used in the exponent of the schedule of the SPARSIFICATION operation. We remind that during a SPARSIFICATION phase, the sparsity varies as sparsity$(t) = s \cdot \frac{1-e^{-ct}}{1-e^{-c}}$ for $t \in [0, 1]$, where $s$ is the target sparsity. The constant $c$ determines how non-linearly the sparsity interpolates from 0 to $s$.

