# OpenReview forum: "SequentialAttention++ for Block Sparsification: Differentiable Pruning Meets Combinatorial Optimization"
_NeurIPS.cc/2024/Conference — NeurIPS 2024 poster_

### Official Review · Reviewer_2z8b · 2024-07-08

**Soundness:** 3
**Presentation:** 3
**Contribution:** 3
**Rating:** 7
**Confidence:** 2

**Summary:**

This paper focuses on bringing a theoretical understanding to differentiable pruning for neural networks and reveals connections to group lasso.

**Strengths:**

Please see the “Questions” section.

**Weaknesses:**

Please see the “Questions” section.

**Questions:**

I think this paper provides some useful connections between optimization theory and pruning. I haven’t checked the math carefully, but things seem reasonable overall. It is interesting that the differential pruning techniques can be explained using group lasso. The empirical results are nice to have, but I view the theoretical connections made in the paper to be the main contribution.

1) Does the training curve in Figure 3 represent the entirety of the training? It looks like for the orange curve, the peak around step=48000 is higher than the accuracy at the end of the training.

2) Also, what happens if the models are trained longer? Will the blue curve continue to do better than the orange curve?

**Limitations:**

Yes.

---

> ### Author Rebuttal · Authors · 2024-08-06
>
> Thank you for the thoughtful review!
>
> > Does the training curve in Figure 3 represent the entirety of the training? It looks like for the orange curve, the peak around step=48000 is higher than the accuracy at the end of the training.
>
> The peak around step=48000 is due to the use of a dense phase during this training step, and is therefore not a fair comparison to the training accuracy of the final sparse model.
>
> > Also, what happens if the models are trained longer? Will the blue curve continue to do better than the orange curve?
>
> We evaluated this by continuing to train both models in that figure with a fixed final learning rate of 1e-3 for another 90 epochs (for a total of 180 epochs). The gap between the two curves does not narrow at all (also the final accuracy of the orange curve (65.83) doesn't reach the accuracy that the blue line had before these extra 90 epochs (67.42)), leading us to conclude that the quality difference between the two algorithms is likely because of the different sets of blocks chosen, rather than a delayed convergence.

---

> > ### Comment · Reviewer_2z8b · 2024-08-12
> >
> > Thank you for the answers.

---

### Official Review · Reviewer_rDbF · 2024-07-11

**Soundness:** 3
**Presentation:** 3
**Contribution:** 3
**Rating:** 6
**Confidence:** 3

**Summary:**

This paper analyzes sparse training with group sparsity and proposes a general theoretical framework to analyze both hard thresholding (discrete) and scoring (continuous) pruning methods. The proposed theoretical framework of block sparsification encompasses multiple existing sparsification methods via non convex regularizers and shows the existence of a unique global minimizer for such problems by establishing an equivalence to the group LASSO problem. Based on the theoretical insights, the proposed algorithm Sequntial Attention++ iteratively prunes parameters combining the approach of AC/DC with block sparsification and score based pruning.

**Strengths:**

1. The paper provides a general theoretical framework to analyze score based sparsification methods and shows the existence of a unique minimizer.
2. The authors build on existing work to propose a new pruning algorithm Sequential Attention++ combining hard thresholding and score based pruning.

**Weaknesses:**

1. The authors discuss that the magnitude is not necessarily the best importance scoring metric for sparsification. However, experimentally it has been observed to be the best performing criteria across multiple datasets for hard thresholding methods with unstructured pruning (see [1]). I would like to know if the authors observe a different trend for (i) structured pruning and (ii) score based sparsification instead of hard thresholding i.e. does magnitude based scoring perform worse in case of differentiable pruning.
2. In the proposed algorithm, the authors use a softmax based scoring method. Does using an unnormalized softmax lead to similar results?
3. If I understand correctly, the proposed results also hold for scoring methods which use the network weights as a score like in STR  [2] or like powerpropagation. However, does the additional overparametrization in case of using a separate scoring parameter improve the overall performance of the network?
4. For ImageNet it would be nice to see comparisons with structured DST [2] as it achieves a state of the art performance for structured sparsity.
5. Have the authors tried the same algorithm without an AC/DC approach, by simply training with a scoring method and pruning after every few epochs. Is the dense phase necessary?

[1] Hoefler, Torsten, et al. "Sparsity in deep learning: Pruning and growth for efficient inference and training in neural networks." Journal of Machine Learning Research (2021).

[2] Kusupati, Aditya, et al. "Soft threshold weight reparameterization for learnable sparsity." International Conference on Machine Learning (2020).

[3] Lasby, Mike, et al. "Dynamic Sparse Training with Structured Sparsity." The Twelfth International Conference on Learning Representations.

**Questions:**

See above.

**Limitations:**

The paper sheds light on the connection between sparsification methods via nonconvex regularizers to the group LASSO problem, thus establishing a theoretical framework to analyze continuous sparsification methods, which is also empirically validated.

---

> ### Author Rebuttal · Authors · 2024-08-06
>
> Thank you for the very detailed review!
>
> > The authors discuss that the magnitude is not necessarily the best importance scoring metric for sparsification. However, experimentally it has been observed to be the best performing criteria across multiple datasets for hard thresholding methods with unstructured pruning (see [1]). I would like to know if the authors observe a different trend for (i) structured pruning and (ii) score based sparsification instead of hard thresholding i.e. does magnitude based scoring perform worse in case of differentiable pruning.
>
> The answer to both (i) and (ii) is yes. For structured (block) pruning, we observe a consistent and strong improvement by using the softmax attention parameterization instead of magnitude scores. In Figure 2a, we compared the quality of one-shot block sparsification when using magnitude scores vs the softmax attention parameterization, and found that the latter gives higher-quality results, especially for larger block sizes (In addition, our results in Tables 1, 2 include baselines like ACDC and one-shot magnitude pruning, which use magnitude scores). For unstructured pruning, it is important to note that the softmax attention reparameterization introduces a large number of additional parameters, reducing its practicality over other methods. Nonetheless, we ran some experiments to answer this question, and found that one-shot softmax attention-based pruning outperforms magnitude pruning even for the setting of unstructured sparsity (accuracy 74.96 vs 74.51 for 80% sparsity and 73.19 vs 72.75 for 90% sparsity). (Technical note for unstructured sparsity: we observed that the quality of the sequential attention reparameterization degrades as the number of candidates becomes too high (as is the case in unstructured sparsity). We mitigated this by decreasing the softmax temperature to 0.25 from the default of 1.0, to obtain the above results).
> More generally, there are also other works that document improvements over magnitude pruning, dating back to the first works on neural network pruning. We have given a discussion of this history along with many references in our introduction.
>
> > In the proposed algorithm, the authors use a softmax based scoring method. Does using an unnormalized softmax lead to similar results?
>
> Based on our experiments, using unnormalized softmax leads to similar but slightly degraded results. For 32x32 block sizes, we have the following results:
>
> ```
> Sparsity | unnormalized | normalized
> 68% 74.69 74.82
> 78% 73.21 73.78
> 88% 69.45 70.82
> 92% (diverged) 65.41
> ```
>
> So experimentally it seems that the normalization provides a small quality gain and some stability improvement. It is possible that these results could be improved by tuning the learning rate and weight decay of the logits used in the exponential function (which together serve as a sort of normalization).
>
>
> > If I understand correctly, the proposed results also hold for scoring methods which use the network weights as a score like in STR [2] or like powerpropagation. However, does the additional overparametrization in case of using a separate scoring parameter improve the overall performance of the network?
>
> Great question. As we remark in Section 2.1.3, applying the Hadamard overparameterization actually increases the concavity (i.e. decreases q) of the equivalent l_q regularizer, which leads to inducing a harder sparsity constraint. As an example, a combination of the Hadamard overparameterization with l1 regularization has been considered in Yang et al. 2019. Quantifying the precise tradeoffs between overparameterization and the choice of non-linear weight activations is an interesting question. As for an experimental comparison, our results on the Criteo dataset in Table 2 include a comparison with Powerpropagation, where we observe a consistent improvement by using SequentialAttention++.
>
> > For ImageNet it would be nice to see comparisons with structured DST [2] as it achieves a state of the art performance for structured sparsity.
>
> We adapted the structured sparsity approach of [2] for block sparsification, however we were not able to produce conclusive results. Fundamentally the STR method requires careful tuning of the weight decay rate and sigmoid initialization in order to properly control the sparsity rate. In order to get a fair comparison, we would have to match the block sparsities in each layer, which is a considerably hard hyperparameter tuning problem. The default hyperparameter values proposed in [2] for 90% sparsity do not seem to transfer to the block sparsity setting.
>
> > Have the authors tried the same algorithm without an AC/DC approach, by simply training with a scoring method and pruning after every few epochs. Is the dense phase necessary?
>
> One challenge with this approach is that, because of weight decay, the scores of the pruned candidate blocks will keep decreasing during the sparse phases. As such, this scheme will be equivalent to one-shot pruning (since the pruned candidate blocks will be too small and will never be selected again after the first pruning). We believe the introduction of the dense phase is important for re-introducing these pruned blocks as active candidates. Something close to this would be to significantly decrease the size of the dense phase. We tried reducing the duration of each dense phase to only around 11 training steps, and let the sparse phases occupy the rest of the training. However, the results are not promising: accuracy 67.28 vs 72.15 for 32x32 blocks and 88% sparsity, likely because of high variance during the short dense phase.

---

> > ### Comment · Reviewer_rDbF · 2024-08-12
> > **Response to rebuttal**
> >
> > Thank you for the detailed response and for answering every question, substantiated with experiments. I appreciate the effort.

---

### Official Review · Reviewer_5Nz3 · 2024-07-13

**Soundness:** 3
**Presentation:** 3
**Contribution:** 4
**Rating:** 6
**Confidence:** 3

**Summary:**

This work studies the task of neural network pruning. The authors unify the two main directions of the literature: differential pruning and combinatorial optimizations. Specifically, they point out that most differentiable pruning techniques can be considered as non-convex regularization group sparse optimization problems. Based on this theoretical analysis, they propose SequentialAttention++. The empirical evaluation shows their method reaching state-of-the-art results.

**Strengths:**

The theoretical insight to unify differential pruning with combinatorial optimizations seems to be a strong theoretical contribution. The empirical evaluation also shows strong performance of the proposed method.

**Weaknesses:**

The writing is slightly hard to follow. Specifically, the exact motivation of SequentialAttention++ seems a bit unclear. Is it just a combination of Sequential Attention and ACDC because they are currently state-of-the-art methods? Or is there a stronger reason and theoretical support behind this?

**Questions:**

Do you have more reasons behind the combination of Sequential Attention and ACDC? Also, can you explain why in most of the experiment results, larger block size seems to perform better? Wouldn't a smaller block size mean more flexibility in pruning?

**Limitations:**

Yes, the authors discuss the limitation of the work.

---

> ### Author Rebuttal · Authors · 2024-08-06
>
> Thank you for taking the time to review our work!
>
> > Is it just a combination of Sequential Attention and ACDC because they are currently state-of-the-art methods? Or is there a stronger reason and theoretical support behind this?
>
> Based on our experiments, as well as the work of Yasuda et al. 2023, the softmax attention parameterization provides a reliable way to rank network components by importance. In particular, for structured (block) sparsity, we find that the softmax attention scores are significantly more reliable than magnitude scores (Figure 2a), especially for larger block sizes. On the other hand, given some proxies for the importance of the candidate components, a very successful approach is to use an iterative method to search over the set of candidates, as opposed to a one-shot pruning approach. For example, orthogonal matching pursuit (OMP) performs forward selection with gradient magnitude scores, IHT/ACDC perform local search with parameter magnitude scores, GMP performs backward selection with parameter magnitude scores, etc. Out of these, ACDC is a state-of-the-art algorithm for neural network pruning, and so we focused on combining it with the softmax attention parameterization as a method that is likely to produce superior results.
>
> > Also, can you explain why in most of the experiment results, larger block size seems to perform better? Wouldn't a smaller block size mean more flexibility in pruning?
>
> If the reviewer is referring to Table 1 (Imagenet results), note that different block sizes are evaluated with different sparsity rates, so it is not true that higher block size leads to better quality. The reason we had to use different sparsity rates is because the layers of ResNet-50 have a wide variety of sizes, and we refrain from sparsifying smaller layers (less than 100 blocks) because it leads to drastic quality degradation. So even though the sparsity rate for layers we sparsify is the same, fewer layers are sparsified as the block size increases. In our results in Table 2 (Criteo dataset) where we only sparsified one layer, increasing the block size leads to worse quality, as expected.

---

> > ### Comment · Reviewer_5Nz3 · 2024-08-11
> >
> > Thank you for answering my questions!

---

### Decision · Program_Chairs · 2024-09-25

**Decision:**

Accept (poster)

**Comment:**

This work addresses structured neural network pruning by unifying differential pruning and combinatorial optimizations. A nonconvex regularization for group sparse optimization is shown to underpin many differentiable pruning methods, and its global optimum is analyzed.  A practical algorithm SequentialAttention++ is then proposed, whose effectiveness is demonstrated on block-wise pruning tasks on the ImageNet and Criteo.

This paper is well written.  It provides new insights on optimization and neural network pruning, and the empirical algorithm is effective.  I am personally a bit concerned that most theoretical results assume the group sparsity is at most 1.  However, overall, the merits of the paper make it a solid contribution to the conference.